# Diagnostic status influences rapport and communicative behaviours in dyadic interactions between autistic and non-autistic people

Themis Nikolas Efthimiou[1]*, Stephanie Lewis[1], Sarah J. Foster[2], Charlotte E. H. Wilks[1], Michelle Dodd[1,3], Lorena Jiménez-Sánchez[1], Danielle Ropar[4], Robert A. Ackerman[2], Noah J. Sasson[2], Sue Fletcher-Watson[5], Catherine J. Crompton[1]

**1** Centre for Clinical Brain Sciences, University of Edinburgh, Edinburgh, United Kingdom, **2** School of Behavioral and Brain Sciences, The University of Texas at Dallas, Richardson, Texas, United States of America, **3** School of Social Sciences, Nottingham Trent University, England, United Kingdom, **4** School of Psychology, University of Nottingham, Nottingham, United Kingdom, **5** Salvesen Mindroom Research Centre, University of Edinburgh, Edinburgh, United Kingdom

* tefthimi@ed.ac.uk

## Abstract

A growing body of research suggests that the behaviours and experiences of autistic and non-autistic people are influenced by whether they are interacting with someone of the same or different diagnostic status. However, little is known about the relationship between these behaviours and the experiences of rapport in matched and mixed neurotype dyads. Using the Actor–Partner Interdependence Model, our pre-registered analyses examine how participants' and their partners' diagnostic statuses influence linguistic, behavioural, and kinematic indices, and how these relate to feelings of rapport among autistic and ($n = 57$; 17 self-diagnosed) non-autistic ($n = 51$) participants interacting within autistic ($n = 20$), non-autistic ($n = 17$), and mixed autistic–non-autistic ($n = 17$) dyads. We found that autistic participants reported lower rapport regardless of their partner's diagnostic status, though awareness of their partner's diagnostic status had a moderating effect. We observed a linguistic difference, autistic participants produced longer mean utterance lengths, these and other behavioural or kinematic indices did not mediate the relationship between diagnostic status and rapport across neurotypes. The current work highlights the need for a nuanced understanding of communication dynamics in autism.

## Introduction

Social interactions are important for developing rapport, defined as a state of harmonious connection and mutual understanding between individuals [1,2]. This connection is developed through the intricate interplay of verbal and nonverbal communication and is crucial for forming social bonds and enhancing well-being [3–9]. Historically, research on social communication has focused on individual

**Data availability statement:** All data are available on the Open Science Framework: https://osf.io/tmuqn/.

**Funding:** Templeton World Charity Foundation, grant number TWCF-2020-20442, which was awarded to CJC, DR, NS, and SF-W. The funders had no role in study design, data collection and analysis, decision to publish, or preparation of the manuscript.

**Competing interests:** The authors have declared that no competing interests exist.

communicative elements like facial expressions, body language, and vocal tone in autistic and non-autistic populations [10–14]. However, these elements rarely occur in isolation during real-world interactions [15,16]. Instead, communication is inherently multimodal, with aspects like laughter seamlessly integrating with facial expressions, vocalisations, and body movements [17,18]. Furthermore, social interactions are dynamic, with adjustments to one person's behaviour occurring in response to their partner [19,20]. Consequently, there has been a methodological shift towards studying dyadic (i.e., two-person) interactions to investigate the structural elements of conversation in non-autistic populations, including head nodding, facial expressivity, turn-taking, conversational share, length and complexity of utterances [21], and gaze patterns, and their influence on conversational enjoyment and workplace success [22–24]. For example, smiles exchanged during conversations can both covertly and causally shape the inferences individuals draw about one another's social intentions [25]. This shift towards understanding communication in naturalistic settings and social contexts is currently understudied in autistic adult populations.

The diagnostic criteria for autism are centred around deficits in social interaction and communication [26–28]. Over recent years, autistic social traits have been increasingly interpreted as being "different" rather than "deficient"; that is, that autistic social communication is a natural variation in interactive styles rather than something that needs to be corrected or mitigated [29]. Additionally, while social cognition research has largely focused on differences in how autistic people produce social signals, for example, reduced simple and complex facial expressions [30], tone of voice [31], verbal and non-verbal feedback [32], and atypical kinematic patterns, such as increased jerk and velocity [27,33,34]. However, many of these findings stem from studies that examine autistic behaviour in isolation or in environments that may not be conducive to their needs and interests [35]. Therefore, a deeper understanding of how autistic people navigate the complexities of communication in real-world settings is essential.

A growing body of research focuses on questions stemming from the Double Empathy Problem (DEP) [36]. The DEP suggests that communication difficulties between autistic and non-autistic groups arise due to a mutual misunderstanding and lack of shared social experiences. This has led to an increased interest in studying autism within dyadic settings, comparing how autistic pairs, non-autistic pairs, and "mixed" pairs (including an autistic and non-autistic person) communicate and build rapport in a shared social context [37]. Findings indicate that information sharing between autistic dyads is on par with non-autistic dyads, with communicative difficulties experienced by mixed dyads [38,39]. Similarly, same-neurotype dyads (i.e., autistic dyads, or non-autistic dyads) experience improved self-rated and observer-rated rapport, compared with mixed dyads [39,40].

While research has begun to examine the success of mixed and matched neurotype interactions in terms of information sharing and/or rapport, few studies have examined how different aspects of communication interact within these contexts. Previous work has shed some light on this area by examining markers of rapport (mutual gaze and backchanneling, i.e., verbal and non-verbal cues like nodding or saying

'uh-huh' to show engagement) in autistic, non-autistic, and mixed pairs [41]. The findings indicated lower rapport ratings and reduced mutual gaze/backchanneling in mixed pairs compared to non-autistic pairs. These differences in communicative indices are not only salient but may also actively interfere with the formation of rapport in mixed dyads. Notably, autistic pairs demonstrated less backchanneling but nevertheless reported higher rapport, suggesting a deviation from the social norms that dictate rapport between non-autistic people. Similarly, a recent study found that facial expression synchrony among autistic children interacting with another autistic child, or neurotypical children interacting with another neurotypical child, predicted enjoyment and a desire for future interaction. This finding further highlights the unique dynamics present in same-neurotype interactions, compared to mixed-neurotype interactions [42]. Together, these studies emphasise the need for deeper investigation into the interactional nuances within mixed and matched neurotype pairs.

However, previous approaches focusing on mixed and matched dyad dynamics do not fully capture the influences each member of the dyad has on the other's outcomes, such as rapport. To address this gap, we utilised the Actor-Partner Interdependence Model (APIM). This statistical model analyses data from dyadic relationships, allowing for the simultaneous examination of how an individual's characteristics influence both their own and their partner's outcomes. This model provides a more nuanced understanding of interpersonal dynamics within the dyad, capturing both individual-level and partner-level influences. Specifically, it distinguishes between actor effects, how an individual's characteristics influence their outcomes, and partner effects, how an individual's characteristics shape their partner's experiences [19,43]. The APIM is particularly advantageous for examining intrapersonal and interpersonal effects within dyadic relationships, including those involving autistic and non-autistic people. The versatility of the APIM has been demonstrated in various contexts, such as romantic partnerships, parent-child relationships, and sibling interactions, where it has been used effectively to analyse dyadic data [44].

In autism research, the APIM has been used to model how rapport varies between autistic and non-autistic adults in both mixed-neurotype and same-neurotype group compositions [45]. This study found that same-neurotype groups (autistic-autistic and non-autistic–non-autistic) reported the highest levels of rapport, particularly in enjoyment, success, and friendliness, compared to mixed-neurotype groups. Autistic participants exhibited a strong preference for interacting with other autistic people, with their reported rapport decreasing as the number of non-autistic participants in the group increased. In contrast, non-autistic participants reported consistent levels of rapport regardless of the group's neurotype composition. As dyadic interactions are the foundational building blocks of group dynamics, the present study focuses on two-person pairings to isolate these core interpersonal behaviours before they are compounded by the complexities of a group setting.

In this study, we examine the potential relationships between actor and partner diagnosis, interactional indices, and self-rated rapport in matched and mixed neurotype pairs. This study is the first to investigate the role of multiple social behaviours, including linguistic (verbal backchannels and utterance length), behavioural (smiles and laughter), and kinematic indices (velocity, acceleration, and jerkiness) and their impact on participant rapport in autistic and non-autistic people.

## Research aims

In this pre-registered paper (https://osf.io/tmuqn/; deviations in Supplemental Material S7 File), we utilised the APIM to analyse how the diagnostic status of both the participant (actor) and their conversation partner (partner) influences self-reported rapport. Additionally, we examined verbal and non-verbal behaviours in conversations, and how these behaviours vary between autistic and non-autistic participants. The study also investigated how these behaviours mediate the relationship with rapport.

Our study examined eight elements of social interactions; each averaged over the 5-minute conversational period. We included three measures of kinematic indices (total upper body velocity, acceleration, and jerkiness [18,27,34]); two measures of emotional expressions (percentage of the conversation spent laughing and smiling [23]); two measures of

verbal interaction (mean utterance length (in seconds) and verbal back-channelling, e.g., "uh-huh," "yeah" [41]); and one measure of non-verbal interaction (non-verbal back-channelling, e.g., nodding). Henceforth, when collectively referring to these variables as a group, we will use the term interactional indices. Our measure of rapport was a self-rated measure that has been applied in previous studies with autistic and non-autistic participants [39,41]. Simply, it is a score out of 500, calculated as the sum of participants' self-reported ratings across five visual analogue scales (0–100), assessing their perception of the interaction in five domains: ease, awkwardness, success, enjoyment, and friendliness (see Supplemental Materials S1 File). We examine how the eight interactional indices may differ based on the diagnostic status of the actors and partners (autistic or non-autistic) and mediate the relationship with rapport.

## Hypotheses

Our hypotheses address the interplay between diagnostic status, interactional indices, and rapport. First (H1a), we hypothesise that self-reported rapport will be lower for autistic participants compared to non-autistic participants, for those paired with an autistic partner compared to those paired with a non-autistic partner (H1b), and in mixed neurotype dyads compared to same-diagnosis dyads (H1c).

Second, we predict that interactional indices will mediate the relationship between diagnostic status and rapport (H2). Specifically, (H2a) we predict that autistic individuals will display reduced behavioural indices (e.g., smiling) and high levels of kinematics indices (i.e., less smooth movements), leading to lower perceived rapport. We also predict a partner effect (H2b), where an actor's rapport is influenced by their partner's behaviour; we hypothesise that an autistic partner will exhibit decreased social behaviours and altered kinematics, which in turn will reduce the actor's perceived rapport. Finally (H2c), we predict that the shared diagnostic status of the dyad moderates these mediation effects. We suggest that the influence of social and kinematic cues on rapport will be strongest in non-autistic dyads and weakest in autistic dyads, with mixed-neurotype dyads falling in between.

Finally, we hypothesise that autistic participants will exhibit diminished behavioural indices and higher kinematic indices, than non-autistic participants (H3a), with these interactional indices further influenced by their partner's diagnostic status (H3b). Finally (H3c), we predict an interaction between the diagnostic statuses of the dyad members will influence interactional indices. Specifically, we hypothesise that autistic participants will exhibit more interactional indices in mixed-neurotype dyads compared to when interacting with an autistic partner. This may be a response to the heightened social demands of masking or adapting their natural interaction style to align with perceived neurotypical norms. Full details and mediation models are provided in S1 File.

## Methods

### Ethics

Ethical approval was obtained from the University of Edinburgh's Medical Research Ethics Committee (21-EMREC-036), the University of Nottingham (F1381), the School of Psychology Ethics Committee, and the University of Texas at Dallas's Institutional Review Board (IRB-21–497). All participants provided written informed consent and were compensated for their time (£30/$40). Data collection occurred across three sites: the University of Dallas (1st September 2022–1st November 2023), the University of Edinburgh (1st September 2022–1st December 2023), and the University of Nottingham (30th of October 2022–1st November 2023).

### Sample size justification

An a priori power analysis was not conducted and the sample size for this study was based on the number of available videos. However, the sample size is consistent with similar dyadic interaction studies [38,39,41]. Furthermore, we conducted a post-hoc sensitivity analysis (see Results: Sensitivity Analysis).

## Participants

The sample comprised 108 participants (autistic: $n = 57$; $M_{age} = 31.04$, $SD_{age} = 13.28$; non-autistic: $n = 51$; $M_{age} = 25.53$, $SD_{age} = 10.26$). Participants were recruited via autism support groups, community centres, via social media, and through networks across three Universities. To confirm that the distribution of dyad types was comparable across the three Universities, a Pearson's Chi-squared test of independence was performed. The analysis showed no significant association between recruitment site and dyad type ($\chi^2(4) = 1.03$, $p = .91$), indicating the groups were sufficiently balanced in this regard.

Participants completed a screening questionnaire administered on Qualtrics to ensure they were over 18, English-speaking, and had normal or corrected vision and hearing. Autistic participants included clinically diagnosed ($n = 38$) and self-identified ($n = 18$), with the self-identified having scored 72 or above on the RAADS-R [46], indicating scores above diagnostic thresholds [47]. In this paper, 'diagnostic status' is used as a collective term to refer to an individual's identification as autistic (either through formal clinical diagnosis or self-diagnosis) or non-autistic. Non-autistic participants scoring over 14 on the RAADS-14 were excluded to maintain low autistic traits [48]. Participants with a social anxiety diagnosis were excluded.

The participants were allocated to one of three dyad conditions: (1) autistic; (2) non-autistic; (3) mixed autistic-non-autistic. Dyad partners were matched semi-randomly based on participant availability on each day of testing, with an aim to minimise gender mismatch where possible. Video quality issues resulted in the exclusion of 12 participants (six dyads), resulting in a sample of 96 participants split into autistic ($n = 47$; $M_{age} = 30.21$, $SD_{age} = 13.53$) and non-autistic ($n = 49$; $M_{age} = 25.53$, $SD_{age} = 10.46$) for verbal and non-verbal behaviour analysis (see Table 1).

## Materials

**Measure of rapport.** Participants' perceived rapport was evaluated using a self-report measure encompassing five dimensions: ease, enjoyment, success, friendliness, and awkwardness (reverse scored). Each dimension was rated on

**Table 1. Participant demographics and dyad distribution for rapport and behavioural analyses.**

| Rapport | Characteristic | Autistic | Mixed | Non-Autistic |
|---|---|---|---|---|
| | | ($n = 20$) | ($n = 17$) | ($n = 17$) |
| | Participants | 40 (11 Self-Diagnosed) | 34 (6 Self- Diagnosed) | 34 |
| | Gender | | | |
| | Male | 6 (15%) | 4 (11.76%) | 10 (29.41%) |
| | Woman | 22 (55%) | 26 (76.47%) | 24 (70.59%) |
| | Non-binary/gender neutral | 11 (27.50%) | 4 (11.76%) | 0 (0.00%) |
| | Prefer not to disclose/self-describe | 1 (2.50%) | 0 (0%) | 0 (0.00%) |
| | Age (years) | 31.98 ± 14.23 | 26.18 ± 9.18 | 26.53 ± 11.65 |
| | IQ-WASI-II | 116.45 ± 15.98 | 112.94 ± 13.01 | 110.06 ± 9.27 |
| Interactional indices | | ($n = 16$) | ($n = 15$) | ($n = 17$)s |
| | Participants | 32 (11 Self- Diagnosed) | 30 5 (Self-Diagnosed) | 34 |
| | Gender | 5 (15.62%) | 3 (10.00%) | 10 (29.41%) |
| | Man | 17 (53.12%) | 24 (80.00%) | 24 (70.59%) |
| | Woman | 10 (31.25%) | 3 (10.00%) | 0 (0.00%) |
| | Non-binary/gender neutral | 0 (0.00%) | 0 (0.00%) | 0 (0.00%) |
| | Age (years) | 31.88 ± 14.89 | 24.97 ± 8.34 | 26.53 ± 11.65 |
| | IQ-WASI-II | 113.94 ± 14.46 | 111.41 ± 12.54 | 110.06 ± 9.27 |

*Note.* This table presents the participant demographics for Rapport Analysis (n = 108) and Behavioural Analysis (n = 96) by group, including average age and standard deviation, along with the number of dyads. Autistic participants and non-autistic participants are organised separately, and mixed dyads include one autistic and one non-autistic participant.

a 100-point visual analogue scale, where higher scores indicated greater rapport (the complete questionnaire and any associated materials presented to participants are available in S2 File). This measure has been previously used in dyadic studies involving both autistic and non-autistic participants [38,39,41]. Additionally, we calculated Cronbach's alpha using the *psych* package [49]. For the entire sample (α = .87) and sample for the social behaviour sample (α = .88), both exceed the recommended threshold of 0.70 [50]. Therefore, a single construct of "rapport" was calculated as the sum of the scores across the five dimensions and used in all subsequent analyses.

### Procedure

We utilised a between-subject design with participant diagnostic status and their partners' diagnostic status (autistic vs non-autistic) as factors. The dyads engaged in social interactions in a controlled setting. Two unfamiliar participants were seated facing each other at an angle between 45° and 90° to facilitate conversation and maintain eye contact if desired. They were instructed to engage in a conversation, with optional prompts provided on a sheet of paper to guide their discussion if needed. Each interaction lasted five minutes and was recorded using Panasonic camcorders (models HC-W580, HC-V785, or HC-V270) mounted on tripods to ensure that both participants were equally visible within the frame. After completing the interaction, participants were separated and asked to complete a self-reported rapport measure (see Fig 1).

### Video processing

Video recordings were analysed using both automated and manual methods to comprehensively assess participant interactions.

For the automated analysis, we used OpenPose for Windows 10 [51–54] to extract 2D (x, y) keypoints for 25 body points of each participant. We focused on upper body keypoints such as the arms (wrist, elbow, shoulder), head (nose), and torso (neck, mid-hip) due to potential occlusion of the lower body. The processing of these data were conducted using the R package *duet* [55], where keypoints with a confidence below 80% were excluded and the remaining data were smoothed, then average velocity, acceleration, and jerk were computed for the entire five minute interaction.

For the manual analysis, video annotation was performed using ELAN software [56], applying a detailed coding scheme that categorised both verbal and non-verbal behaviours (the complete coding scheme is provided in S6 File).

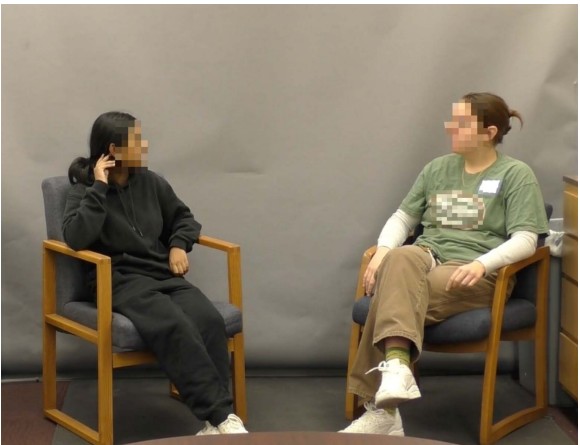

**Fig 1. Study setup for dyadic social interaction.** Note. Example layout of the study setup for the two members of a single dyad engaged in a social interaction. The image illustrates the spatial arrangement and environment used in the study, with participants' faces blurred out to ensure anonymity.

Coded behaviours included shortest and longest utterances, mean utterance length in seconds, number of verbal and non-verbal backchannels, and the frequency and mean durations of laughs and smiles. To ensure accuracy and consistency, inter-rater reliability was established by having multiple coders independently annotate the same video segments, and agreement rates were calculated to confirm the reliability of the coded data.

Inter-rater reliability for five variables was assessed using the Intraclass Correlation Coefficient (ICC) to evaluate the degree of agreement between two raters. Specifically, a two-way random-effects model was applied to obtain a single-measure ICC under the absolute agreement criterion [57]. Excellent reliability was observed for Mean Utterance Length (.96), Percent Laughing (.93), Verbal Backchanneling Rate (.91), and Nonverbal Backchanneling Rate (0.95), indicating strong consistency across these measures. However, the variable Percent Smiling showed a notably lower ICC (.31), reflecting weaker agreement between raters.

## Data analysis

All data were analysed using R version 4.3.2 [58] and scripts are available on the Open Science Framework (https://osf.io/tmuqn/). Multilevel modelling was carried out using the *lme4* [59] and *lmerTest* [60] packages to compute p-values and degrees of freedom [60]. The mediation model was carried out using the *lavaan* package [61]. All models were assessed for multicollinearity, normality of residuals, and appropriate degrees of freedom. Robust linear mixed models of the *robustlmm* package [62] and analyses using the Satterthwaite [60] were carried out in conjunction to ensure the robustness of the models. Post-hoc comparisons were carried out using the emmeans package [63]. All analyses used the conventional alpha level of .05, and for the registered exploratory analysis, p-values were corrected for multiple comparisons using the Bonferroni correction. Further, for each variable, we conducted a linear mixed-effects model with contrasts set to sum-to-zero coding (contr.sum), and reported the results using Type III ANOVA outputs.

To further assess the robustness of our findings (H1, H2, H3), we fitted equivalent Bayesian models using alternative family distributions. Although these models produced somewhat different effect sizes, they largely converged on the same overall conclusions (full details of the Bayesian model specifications, priors, and outputs are located in S5 File).

## Data transformations

To implement the APIM, we effects-coded the diagnostic status of both actors and their partners using the *car* package [64]. Specifically, we transformed the original variables for actors and partners into binary indicators representing autistic and non-autistic statuses. For each participant in the dyad, two separate dummy variables were created: one indicating whether the participant was autistic and another indicating if they were not autistic. This involved recoding the original variables so that a value of 1 denoted an autistic status and −1 denoted a non-autistic status for both actors and partners.

## Results

For H1, we used the APIM to estimate the effects of participants' and their partners' diagnostic status on participants' reported rapport. The average reported rapport was 391.90 (out of 500).

Consistent with H1a, we found a significant actor effect for diagnostic status ($b = -31.43$, $SE = 7.97$, 95% CI [−47.05 −15.81], $p < .001$; see Fig 2), such that autistic participants reported significantly lower levels of overall rapport ($M = 360.47$, $SE = 12.91$) compared to non-autistic participants ($b = 423.32$, $SE = 13.34$). Contrary to H1b, the diagnostic status of the partner did not result in significantly different levels of overall rapport ($b = 7.75$, $SE = 7.97$, 95% CI [−7.87 23.37], $p = .33$). Additionally, H1c was not supported; the effect of participants' diagnostic status on the overall level of rapport did not significantly depend on whether their interaction partner shared the same diagnostic status ($b = -2.54$, $SE = 10.43$, 95% CI [−22.99 17.90], $p = .81$). These findings indicate that an individual's diagnostic status is a significant predictor of their perceived rapport, with autistic participants reporting lower rapport. We did not observe a partner or actor-partner interaction effects.

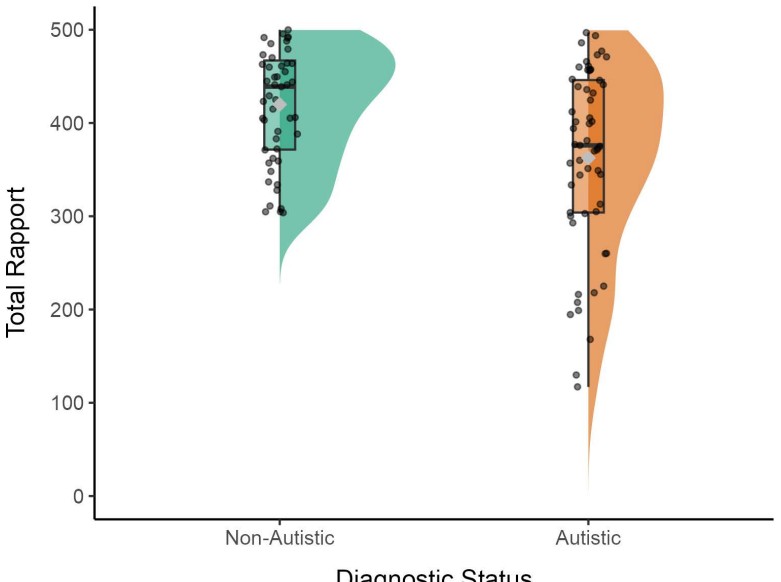

**Fig 2. Visualisation of the total rapport scores for actors across diagnostic statuses.** Note. The half-violin plots show the distribution of total rapport scores, overlaid with boxplots summarising the median and interquartile ranges, and jittered points representing individual data. The grey diamond indicates the mean for each diagnostic group. Non-autistic and autistic diagnostic statuses are shown, highlighting differences in rapport across actor roles.

We examined the mediation hypotheses (H2a, H2b, and H2c) to determine whether social behaviours and kinematic indices mediated the relationship between diagnostic status and reported rapport. Specifically, Hypothesis 2a posited that participants' interactional indices (verbal backchannel rate, nonverbal backchannel rate, percent laughing, percent smiling, and mean utterance length, velocity, acceleration, and jerkiness) would mediate the association between their diagnostic status and overall rapport. H2b suggested that partners' social behaviours would mediate the relationship between partners' diagnostic status and participants' rapport. Additionally, Hypothesis 2c proposed that the mediation effects would differ based on whether dyads shared the same diagnostic status.

To systematically assess these mediation pathways, we conducted separate SEMs for each mediating variable. The indirect effects of each mediator on the relationship between diagnostic status and rapport are summarised in Table 2.

Overall, we did not confirm H2 as the mediation analyses indicated that neither participants' nor their partners' interactional indices significantly mediated the relationship between diagnostic status and perceived rapport.

We tested H3, which examined whether participants' and partners' diagnostic statuses predicted various interactional indices. The results for each behaviour are summarised in Table 3. We observed differences in mean utterance length, specifically autistic participants exhibited significantly longer mean utterance lengths compared to non-autistic participants ($b = 1.46$, $SE = 0.34$, 95% CI [0.79, 2.13], $p < .001$), suggesting that, on average, autistic participants' utterances were 1.46 seconds longer per turn than those of non-autistic participants.

### Registered exploratory analyses

We registered several exploratory analyses, aggregated dyad-level interactional indices (S3 File) and a more granular breakdown of kinematic indices by joint (S4 File), neither of which resulted in significant effects. For our investigation of the effect of awareness of partners' diagnostic status on rapport, an ANOVA revealed the main effect of neurotype was statistically significant ($F(2, 49.52) = 3.40$, $p = .041$, $η² = 0.12$). Post hoc comparisons, adjusted for multiple comparisons,

**Table 2. Indirect effects of social behaviours and kinematic indices on the relationship between diagnostic status and rapport.**

| Hypothesis | b | SE | p | 95% CI |
|---|---|---|---|---|
| H2a | | | | |
| Verbal Backchannels | 1.02 | 2.85 | .72 | [−4.56 6.59] |
| Nonverbal Backchannels | −2.01 | 2.03 | .32 | [−5.99 1.96] |
| Percent Laughing | −0.07 | 0.42 | .87 | [−0.88 0.75] |
| Percent Smiling | 0.24 | 1.153 | .84 | [−2.02 2.50] |
| Mean Utterance Length | 3.27 | 3.822 | .39 | [−4.23 10.76] |
| Acceleration | 0.02 | 0.34 | .95 | [−0.65 0.69] |
| Velocity | 0.18 | 0.651 | .78 | [−1.10 1.46] |
| Jerkiness | 0.09 | 0.463 | .85 | [−0.821 0.99] |
| H2b | | | | |
| Verbal Backchannels | 2.72 | 2.99 | .35 | [−3.12 8.60] |
| Nonverbal Backchannels | −0.92 | 1.56 | .55 | [−3.98 2.14] |
| Percent Laughing | −0.30 | 0.75 | .69 | [−1.76 1.17] |
| Percent Smiling | −2.22 | 2.04 | .27 | [−6.22 1.78 |
| Mean Utterance Length | 5.83 | 3.97 | .14 | [−1.955 13.61] |
| Acceleration | −0.15 | 0.51 | .77 | [−1.144 0.84] |
| Velocity | 0.18 | 0.65 | .78 | [−1.097 1.46] |
| Jerkiness | −0.11 | 0.49 | .82 | [−1.075 0.85] |
| H2c (Actor-Actor Pathway) | 0.26 | 0.97 | .79 | [−1.65 2.17] |
| Verbal Backchannels | −2.33 | 2.93 | .43 | [−8.07 3.40] |
| Nonverbal Backchannels | −0.54 | 3.21 | .87 | [−6.83 5.75] |
| Percent Laughing | 0.53 | 2.60 | .84 | [−4.572 5.64] |
| Percent Smiling | −0.01 | 1.66 | .99 | [−3.26 3.25] |
| Mean Utterance Length | 5.83 | 3.97 | .14 | [−1.955 13.61] |
| Acceleration | 0.24 | 4.12 | .06 | [−7.84 8.32] |
| Velocity | −2.84 | 4.24 | .50 | [−11.15 5.46] |
| Jerkiness | 0.72 | 3.57 | .84 | [−6.27 7.70] |
| H2c (Actor-Partner Pathway) | | | | |
| Verbal Backchannels | 0.80 | 1.87 | .71 | [−2.97 4.37] |
| Nonverbal Backchannels | −1.07 | 1.95 | .59 | [−4.90 2.76] |
| Percent Laughing | −2.40 | 3.53 | .50 | [−9.33 4.52] |
| Percent Smiling | −4.96 | 5.50 | .37 | [−15.74 5.83] |
| Mean Utterance Length | −0.01 | 2.96 | .99 | [−5.82 5.80] |
| Acceleration | −1.82 | 4.21 | .67 | [−10.07 6.43] |
| Velocity | −2.86 | 4.24 | .50 | [−11.17 5.45] |
| Jerkiness | −0.95 | 3.56 | .79 | [−7.928 6.03] |

*Note.* The table shows the indirect effects of various social behaviours and kinematic indices on the relationship between diagnostic status and rapport. None of the indirect effects for H2a, H2b, and H2c reached statistical significance (p-values > .05).

of the marginal means revealed that non-autistic dyads reported significantly higher rapport scores than autistic dyads ($M_{diff}$ = 49.90, SE = 21.9, $t(50)$ = 2.28, p = .027, p-adjusted = .08, Cohen's d = 0.72) and mixed groups had significantly higher rapport scores than the autistic group ($M_{diff}$ = 45.22, SE = 21.4, $t(48)$ = 2.12, p = .039, $p_{adjusted}$ = .12, Cohen's d = 0.65). There was no significant difference between non-autistic and mixed pairs ($M_{diff}$ = 4.67, SE = 23.1, $t(49.8)$ = 0.20, p = .84, $p_{adjusted}$ = 1.000, Cohens d = 0.07).

**Table 3. APIM results for hypothesis 3.**

| Outcome Variable | Predictor | b | SE | 95% CI | p |
|---|---|---|---|---|---|
| Verbal Backchannel Rate | Actor (H3a) | −0.02 | 0.01 | [−0.03 0.00] | .07 |
| | Partner (H3b) | 0.002 | 0.01 | [−0.01 0.02] | .83 |
| | Interaction (H3c) | −0.002 | 0.01 | [−0.02 0.02] | .84 |
| Nonverbal Backchannel Rate | Actor (H3a) | 0.01 | 0.01 | [−0.01 0.02] | .50 |
| | Partner (H3b) | −0.001 | 0.01 | [−0.02 0.01] | .91 |
| | Interaction (H3c) | −0.01 | 0.01 | [−0.03 0.01] | .33 |
| Percent Laughing | Actor(H3a) | −0.18 | 0.40 | [−0.97 0.61] | .65 |
| | Partner (H3b) | −0.46 | 0.40 | [−1.25 0.33] | .26 |
| | Interaction (H3c) | −0.76 | 0.51 | [−1.76 0.24] | .14 |
| Percent Smiling | Actor(H3a) | 2.65 | 3.07 | [−1.44 6.74] | .21 |
| | Partner (H3b) | 1.02 | 2.09 | [−3.07 5.11] | .63 |
| | Interaction (H3c) | 3.14 | 2.09 | [−2.87 9.15] | .31 |
| Mean Utterance Length | Actor (H3a) | 1.46 | 0.34 | [0.79 2.13] | <.001 |
| | Partner (H3b) | −0.32 | 0.34 | [−0.99 0.35] | .35 |
| | Interaction (H3c) | −0.02 | 0.38 | [−0.77 0.73] | .95 |
| Acceleration | Actor(H3a) | −0.93 | 2.39 | [−5.62 3.76] | .70 |
| | Partner (H3b) | 0.83 | 2.39 | [−3.86 5.51] | .73 |
| | Interaction (H3c) | −5.71 | 3.21 | [−12.01 0.59] | .08 |
| Velocity | Actor(H3a) | 0.20 | 0.68 | [−1.14 1.54] | .77 |
| | Partner (H3b) | −0.30 | 0.68 | [−1.63 1.04] | .66 |
| | Interaction (H3c) | −1.62 | 0.86 | [−3.31 0.08] | .06 |
| Jerkiness | Actor(H3a) | −39.18 | 80.58 | [−197.12 118.76] | .63 |
| | Partner (H3b) | 55.03 | 80.58 | [−102.91 212.97] | .50 |
| | Interaction (H3c) | −165.91 | 109.21 | [−379.96 48.14] | .13 |

*Note.* The table presents the estimated effects of actor diagnostic status on various behaviours. The beta coefficients represent the direction and magnitude of the difference between autistic (coded as 1) and non-autistic (coded as −1) participants in each model. Positive beta values indicate that autistic participants exhibit higher values for the respective behaviour compared to non-autistic participants, while negative values suggest lower values. All models include dyad ID as a random effect to account for shared variance within dyads. Grey shaded cells denote statistical significance (p < .05).

Furthermore, the interaction between neurotype and blinding was statistically significant ($F(2, 55.37) = 8.09$, $p < .001$, $\eta^2 = 0.23$). To deconstruct this interaction, we examined the simple effects of neurotype within each blinding condition and the simple effects of blinding within each neurotype.

When dyads were informed of their partner's diagnostic status, there were no significant differences in rapport between any of the neurotype groups (all $ts < 1.91$, $p_{adjusted} > .18$). In contrast, when dyads were uninformed, significant differences emerged. Specifically, uninformed autistic dyads overall rapport was lower compared to mixed ($M_{diff} = 90.38$, $t(48.0) = 3.95$, $p_{adjusted} < .001$, Cohen's $d = 0.13$) and uninformed non-autistic dyads ($M_{diff} = 131.08$, $t(57.8) = 2.61$, $p_{adjusted} = .035$, Cohen's $d = 0.19$). In summary, when dyads were informed of each other's diagnostic status, no significant differences in rapport were found between neurotypes. However, when uninformed, autistic dyads consistently reported lower rapport.

Alternatively, when conditioned by neurotype (see Fig 3 and Table 4 for comparisons), we observe that uninformed mixed dyads report higher rapport than the informed mixed dyads, whereas the opposite is true for the autistic dyads.

It should be noted that, descriptively, the differences in the mixed neurotype dyads appear to be driven by the autistic member (Informed: $n = 22$, $M = 365.82$, $SD = 92.03$; Uninformed: $n = 12$, $M = 447$, $SD = 46.2$). In other words, when autistic people are interacting with a non-autistic person, not knowing each other's diagnostic status results in a more positive

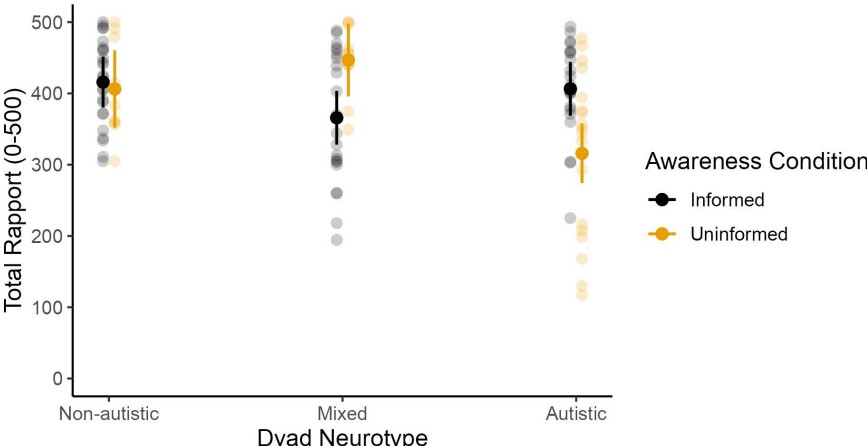

**Fig 3. Dyad neurotype and partner's diagnostic status affect rapport.** *Note*. Transparent data points represent individual observations, and solid points and lines represent model-predicted averages with 95% confidence intervals.

**Table 4. Post-hoc comparisons by neurotype.**

| Neurotype | Contrast | *Mdiff* | *SE* | *df* | *t* | *p* | *Cohens d* |
|-----------|----------|---------|------|------|-----|-----|------------|
| Non-autistic | Informed – Uninformed | 9.66 | 32.60 | 72.8 | 0.30 | .76 | 0.13 |
| Mixed | Informed – Uninformed | −81.1 | 32.00 | 48 | −2.54 | .015 | −1.16 |
| Autistic | Informed – Uninformed | 90.62 | 28.30 | 48 | 3.20 | .002 | 1.30 |

*Note.* Post hoc comparisons for the interaction effect show the differences in rapport scores between the Informed and Uninformed conditions for each neurotype, along with Bonferroni-adjusted p-values.

experience for the autistic partner. But when two autistic people are interacting, they have the greatest rapport when they are aware their partner is autistic.

To examine whether the neurotype of an interaction partner influenced how autistic participants rated their experience, a Welch independent-samples t-test was conducted. This compared the self-reported rapport of autistic participants who interacted with another autistic person ($M = 365.68$) versus those who interacted with a non-autistic person ($M = 355.26$). The results indicated no significant difference between the two groups, $t(33.02) = 0.38$, $p = .709$. Consistent with our APIM model, these results suggest that autistic participants' rapport ratings were not dependent on their partner's neurotype in this study.

## Sensitivity analysis

We conducted an effect-size sensitivity analysis for the APIM using the online APIM Power Analysis application by Ackerman and Kenny (2024; https://robert-a-ackerman.shinyapps.io/APIMPowerRdis/). This tool provided power estimates for detecting actor and partner effects in indistinguishable dyads — that is, dyads in which there is no meaningful variable with which to tell the two dyad members apart— with our sample size and specific parameters. We set the effect size (Cohen's *d*) for the actor effect at 0.55 and the partner effect at 0.50, with a correlation of 0 between the actor and partner variables and an error correlation of 0.5. With a sample size of 54 dyads and an alpha level of .05, we obtained a power of .80 to detect an actor effect of size 0.55 ($\beta = .26$, partial $r = .267$, ncp $= 2.84$) and a power of 0.72 to detect a partner effect of size 0.50 ($\beta = .24$, partial $r = .245$, ncp $= 2.58$). These results indicate that we have sufficient power to detect moderate actor and partner effects in our model, with greater power observed for the actor effect. Our current sample size suggests

that we are likely to identify meaningful actor effects, where an individual's behaviour affects their outcomes. However, we have slightly lower power for detecting partner effects, where an individual's behaviour impacts their partner's outcomes, which could require a larger sample to achieve equivalent power.

## Discussion

In this study, we investigated how diagnostic status (autistic or non-autistic) affects self-reported rapport and specific communicative behaviours in dyadic interactions. Autistic participants generally reported lower rapport, supporting our first hypothesis, regardless of their partner's diagnostic status. SEMs showed no behaviours or kinematic indices that might mediate the relationship between diagnostic status and rapport. Consequently, the underlying mechanisms contributing to the lower reported rapport among autistic participants remain unclear, suggesting that additional unmeasured factors or alternative processes may be involved.

Our analysis of individual behaviours (H3) showed that autistic participants exhibited longer mean utterance lengths than non-autistic participants—a finding that contradicts our initial prediction. No other significant differences were observed, a result that diverges from prior research reporting generally reduced social behaviours and kinematics in autistic people [27,65,66]. These outcomes underscore the complexity of social communication and indicate that further investigation is required to elucidate the processes influencing rapport and communicative patterns.

Our findings show that diagnostic status significantly influences self-reported rapport, consistent with prior research [45]. Autistic participants reported lower rapport, potentially due to heightened anxiety during the unstructured interaction [67]. Indeed, the pre-registered exploratory analysis, at the dyadic level, revealed that awareness of the partner's diagnostic status moderated rapport: autistic participants reported lower rapport and verbal backchanneling when unaware their partner was non-autistic. Conversely, rapport and backchanneling increased when they knew they were interacting with an autistic person. These findings align with studies highlighting positive experiences among autistic peers [68–70], as they have distinct modes of social interaction that work well among themselves, similar to interaction style or movement synchronisation [39,71]. It is noteworthy that this effect was observed exclusively in the informed condition. One interpretation is that having explicit information about the partner's diagnostic status may eliminate the need for participants to attempt to infer this information, thereby potentially enhancing the quality of the interaction.

Although we were unable to identify potential mechanisms for these distinct forms of social interactions, our mediation analysis revealed no verbal, non-verbal, or kinematic indices mediating the relationship between diagnostic status and rapport. This result contrasts with previous research, which has shown that autistic pairs produce less verbal backchanneling—a reduction that has been associated with higher ratings of rapport [41,72]. This may mean that autistic people engage more deeply by speaking more, or they enjoy interactions and thus contribute more verbally [73], fostering stronger connections. Future research may replicate this approach to investigate the directionality and underlying mechanisms to confirm the mediation effects, while attempting to establish causality [25].

Notably, our study observed differences between autistic and non-autistic participants only in linguistic elements, specifically, mean utterance length, while no significant differences were found in the other interactional indices. This may be attributable to the nature of the task: an unstructured seated conversation that inherently relies on verbal exchange rather than movement and gestures. The experimental setting might have been perceived as formal, as participants were meeting a new person within the context of a study, and people may use fewer gestures and exhibit more restrained body language in formal situations [74]. Those who are less comfortable in such settings may be particularly inclined to minimise their use of gestures and nonverbal expressions. Furthermore, our study captured only a limited range of social and kinematic behaviours during the interaction and did not account for other elements that could influence the dynamics of the exchange, such as subtle gestures, postural adjustments, or contextual factors that may emerge over longer periods. Therefore, the absence of observed differences in motion or expressions may reflect both the task's emphasis on verbal communication and the constraints of our measurement approach. This suggests that future research should consider

employing extended interaction periods and more comprehensive assessments of nonverbal behaviours to fully understand their role in social interactions between autistic and non-autistic people.

Despite the insights provided by this study, several methodological considerations and limitations warrant acknowledgement. First, while self-reported measures of rapport are the most direct way to assess how a participant has experienced an interaction, other factors may affect responses. Internalised stigma or a reduced tendency to provide socially desirable responses may contribute to lower self-reported rapport among autistic participants [75–77]. Autistic people may also experience alexithymia, a difficulty in identifying and describing one's emotions [78,79], which may also affect responses; however, the relationship between alexithymia and social perception is currently unclear. Second, as indicated by our sensitivity analysis, the study may have been underpowered for detecting partner effects (H1b) and interaction effects (H1c) within the Actor-Partner Interdependence Model, suggesting that these non-significant findings should be interpreted with caution. Thirdly, it should also be noted that the behavioural coding for smiling, while reliable, was subject to some ambiguity; suboptimal camera angles at times resulted in only partial facial visibility, and coders occasionally faced challenges in differentiating subtle smiles from laughter, which may have introduced noise into this specific measure. Fourth, our participants had relative high IQs, and as such may not be representative of the wider population. Fifth, speech was quantified using mean utterance length in seconds. While this is a useful measure of conversational share and dynamics, we did not examine the linguistic content of what participants discussed, which may have influenced their rapport development. Finally, the analysis treated each communicative behaviour in isolation, potentially overlooking the dynamic interplay between different behaviours that characterise real-world social interactions. Future research should consider approaches that capture these interdependencies to better understand the complex mechanisms underlying rapport.

Future research should address these limitations by incorporating larger and more diverse samples to increase statistical power, particularly for detecting partner and interaction effects. Given that our primary differences were linguistic, notably in mean utterance length and a marginal effect for verbal backchannels, subsequent studies should examine language factors in greater depth. This may include looking specifically at the linguistic content of the interactions at the semantic and syntactic level, as well as exploring paralinguistic features such as prosody and pitch, which may reveal whether more subtle linguistic variations contribute to differences in rapport [80]. Additionally, employing autistic-friendly environments and methodologies, such as longer interaction periods, interactions requiring increased motion and structured tasks [71], or settings [81] that accommodate sensory sensitivities, may enhance the validity of the findings. Furthermore, examining communicative behaviours in conjunction could provide a more holistic understanding of how these factors collectively impact social rapport. Incorporating objective measures alongside self-reports, such as behavioural observations or physiological indicators, may also help mitigate potential biases associated with self-assessment. By addressing these areas, future studies can build upon the current findings to develop more effective interventions aimed at improving social interactions for autistic participants.

## Supporting information

**S1 File. Detailed hypotheses and mediation models.**
(DOCX)

**S2 File. Rapport measurement materials.** Materials presented to participants to measure rapport.
(DOCX)

**S3 File. Dyad-level analysis.** Dyad-level analysis on verbal/non-verbal and kinematic indices.
(DOCX)

**S4 File. Joint-level kinematic analysis.** Analysis of kinematic indices by joint.
(DOCX)

**S5 File. Bayesian model details.** Details on analogous Bayesian models.
(DOCX)

**S6 File. Complete coding scheme.**
(DOCX)

**S7 File. Deviations from pre-registration.**
(DOCX)

## Author contributions

**Conceptualization:** Themis Nikolas Efthimiou, Robert A. Ackerman, Noah J. Sasson, Catherine J. Crompton.

**Data curation:** Themis Nikolas Efthimiou.

**Formal analysis:** Themis Nikolas Efthimiou, Robert A. Ackerman.

**Funding acquisition:** Danielle Ropar, Noah J. Sasson, Sue Fletcher-Watson.

**Investigation:** Stephanie Lewis, Sarah J. Foster, Charlotte E.H. Wilks, Michelle Dodd.

**Methodology:** Catherine J. Crompton.

**Project administration:** Catherine J. Crompton.

**Resources:** Lorena Jiménez-Sánchez.

**Supervision:** Catherine J. Crompton.

**Validation:** Lorena Jiménez-Sánchez.

**Visualization:** Themis Nikolas Efthimiou.

**Writing – original draft:** Themis Nikolas Efthimiou.

**Writing – review & editing:** Stephanie Lewis, Sarah J. Foster, Charlotte E.H. Wilks, Michelle Dodd, Lorena Jiménez-Sánchez, Danielle Ropar, Robert A. Ackerman, Noah J. Sasson, Sue Fletcher-Watson, Catherine J. Crompton.

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
