## [Decision Letter · Decision Letter 0]

16 Jun 2025

PONE-D-25-10256Diagnostic status influences rapport and communicative behaviours in dyadic interactions between autistic and non-autistic people

PLOS ONE

Dear Dr. Efthimiou,

Thank you for submitting your manuscript to PLOS ONE. After careful consideration, we feel that it has merit but does not fully meet PLOS ONE’s publication criteria as it currently stands. Therefore, we invite you to submit a revised version of the manuscript that addresses the points raised during the review process.

After careful analysis, both the reviewers and I agree that your paper addresses a significant issue, presents a novel and attractive research approach, and contributes to our understanding of communication behaviors among autistic individuals in dyadic interactions. Overall, the manuscript is well-written. However, several aspects require further refinement and clarification. These include details regarding the research methodology, such as the formulation of hypotheses, the rationale behind the choice of indices, and a description of the process for matching dyad partners, as well as certain unclear sections in the introduction and discussion, as highlighted in the reviews. I believe that addressing the reviewers’ insightful comments will improve the quality of the manuscript.

We look forward to receiving your revised manuscript.

Kind regards,

Ewa Pisula

Academic Editor

PLOS ONE

Journal Requirements:

2. Please describe in your methods section how capacity to provide consent was determined for the participants in this study. Please also state whether your ethics committee or IRB approved this consent procedure. If you did not assess capacity to consent please briefly outline why this was not necessary in this case.

“Templeton World Charity Foundation, grant number TWCF-2020-20442, which was awarded to CJC, DR, NS, and SF-W”

5. We note that Figure 1 in your submission contain copyrighted image. All PLOS content is published under the Creative Commons Attribution License (CC BY 4.0), which means that the manuscript, images, and Supporting Information files will be freely available online, and any third party is permitted to access, download, copy, distribute, and use these materials in any way, even commercially, with proper attribution. For more information, see our copyright guidelines: http://journals.plos.org/plosone/s/licenses-and-copyright.

6. We note you have included a table to which you do not refer in the text of your manuscript. Please ensure that you refer to Table 4 in your text; if accepted, production will need this reference to link the reader to the Table.

8. Please remove all personal information, ensure that the data shared are in accordance with participant consent, and re-upload a fully anonymized data set.

Additional Editor Comments:

After careful analysis, both the reviewers and I agree that your paper addresses a significant issue, presents a novel and attractive research approach, and contributes to our understanding of communication behaviors among autistic individuals in dyadic interactions. Overall, the manuscript is well-written. However, several aspects require further refinement and clarification. These include details regarding the research methodology, such as the formulation of hypotheses, the rationale behind the choice of indices, and a description of the process for matching dyad partners, as well as certain unclear sections in the introduction and discussion, as highlighted in the reviews. I believe that addressing the reviewers’ insightful comments will improve the quality of the manuscript.

Reviewers' comments:

Reviewer's Responses to Questions

**Comments to the Author**

1. Is the manuscript technically sound, and do the data support the conclusions?

Reviewer #1: Yes

Reviewer #2: Yes

2. Has the statistical analysis been performed appropriately and rigorously? 

Reviewer #1: Yes

Reviewer #2: Yes

3. Have the authors made all data underlying the findings in their manuscript fully available?

Reviewer #1: No

Reviewer #2: Yes

4. Is the manuscript presented in an intelligible fashion and written in standard English?

Reviewer #1: Yes

Reviewer #2: Yes

5. Review Comments to the Author

Reviewer #1: The manuscript in question tackles an intriguing question using a novel and appropriate approach. It was also enjoyable to read. The authors are correct that this area is currently understudied and I believe their report will present a valuable addition to the study of Autism and Social Cognition. Nonetheless, I have a few thoughts I would like to offer for consideration before I can recommend this article for publication.

The introduction is very informative and focused, but I'd like to see a brief explanation/definition of the terms rapport and backchanneling before these are used in the argument - the former is sometimes ambiguous and can be operationalised in different ways, whereas the latter may not be known to many readers. The role of kinematics and why they matter when exploring rapport and comunication could also be established more as that is another non-intuitive factor.

I would also appreciate if the authors could briefly cover the deficits-or-differences argument, as this is very relevant here. The authors already bring up the current implicit understanding of Autism in terms of social deficits (line 59) that contrasts with the cited literature showing effective communication and high rapport among matched pairs. I would just like to see this addressed explciitly.

The authors mention larger group setting (line 92) and groups larger than dyads (line 114) - briefly addressing dyadic vs group interactions may be informative and clarify the scope of the current manuscript.

For understandable reasons, the introduction and article in general treat "Autistic" and "Non-Autistic" as homogenous groups. It is okay and expected if intra-group diversity is not within scope, but this is a broad issue within the field at large so it should be explicitly stated as a limitation: Both co-occuring conditions, and different levels of Autism (certainly level 3) - would no-doubt affect the results and dyadic communication, as would other individual differences and conditions among the non-Autism group that were not recorded.

I notice that participants were recruited from different countries. Could the authors please clarify whether each country contributed the sae proportion to each dyad-arrangement? It doesn't have to be equal between countries but it should be equal across arrangements. If any country contributed, for example, a majority of Autistic dyads but a minority of Non-Autistic dyads, then cultural factors become an important consideration.

Might the results regarding the diagnostic status of the partner be influenced by social desirability? It seems logical that one dyad member would be hesitant to report low rapport if they knew their partner was Autistic, which would explain the high rapport of mixed dyads. This would be an interesting effect in itself but would present a confound that should be mentioned. To further explore this, the authors could compare the rapport ratings within mixed pairs, looking for differencxes between the Autistic and non-Autistic partner's ratings.

I may be reading this wrong (which may affect my previous point), but in the paragraph starting on line 362, the authors seem to say that informed mixed dyads report higher rapport, which is driven by the Autistic member who, when informed, reports lower rapport. This is obviously contradictory so I am guessing something may have gotten switched around here or the paragraph is not entirely clear. Upon reviewing figure 3 and table 4, I think there is a mistake in line 363, which possibly should say that uninformed mixed dyads reported higher rapport, not lower. In that case my previous point becomes moot but I would still like to see a deeper discussion of this effect.

It is already part of the data but a direct test comparing ratings of Autistic participants, depending on whether they interacted with an Autistic or non-Autistic partner, may be interesting. Currently this is somewhat hard to judge due to mixed pairs being averaged across the two participants.

Minor comments:

On line 96, the words "is a" are repeated.

Hypotheses H2b (line 152) and H3c (line 160) are phrased in a way that is very hard to parse. Please consider rewriting.

In hypothesis H3a (line 158), I believe the authors mean to say that Autistic participants will exhibit dimished or reduced multimodal indices, as the amount of indices stays the same.

Errant ")" on line 190.

Reviewer #2: Thank you for the opportunity to review this article. This is a well-written paper presenting a thoughtfully designed study that makes a valuable contribution to the field. The comprehensive analysis of verbal and non-verbal cues across same- and mixed-neurotype interactions addresses an important gap in the existing literature. I offer the following suggestions to enhance the paper's clarity.

Page 3, line 57: I find this sentence unclear. If a shift is mentioned, I would expect some contrast between views/methodologies to be present in the previous sentences, which is not the case.

Page 6, lines 132-133: This sentence is slightly misleading. When first reading it, I assumed that utterance length had been measured in number of words, since it is introduced as a verbal interaction metric. However, the supplementary material reveals that length was measured in seconds, including silent pauses—which constitutes a non-verbal component. To avoid confusion, I suggest reformulating this sentence and/or specifying upfront that utterance length refers to the temporal duration of utterances.

I would also like to see a brief explanation of 1) why utterance length was chosen as an interaction metric (utterance length is not mentioned in the Introduction, whereas other measures such as smiles, nods, movements are) and 2) why the authors chose to measure it in seconds rather than in words ?

Page 6, line 136: I also wonder if referring to the eight elements of social interaction as multimodal indices is correct since most elements are not multimodal but unimodal (e.g., nods, smiling, total upper body velocity, acceleration, and jerkiness).

Page 7, lines 144-148: I find the formulation of hypothesis 1(and its sub-hypotheses) unclear and would suggest reformulating as separate sentences or as bullet points. Furthermore, it is not clear from the introduction and literature review why the authors hypothesise that self-reported rapport will be lower for autistic participants compared to non-autistic participants.

Page 7, lines 149-164: Hypothesis 2a mentions social behaviors, are those the same as multimodal indices mentioned in Hypothesis 3? If yes, why use two different terms? If not, what does social behaviors refer to?

Likewise, Hypothesis 2a stipulates that autistic individuals will produce increased, less smooth movements but Hypothesis 3a stipulates that they will exhibit fewer multimodal indices (which includes kinematic measures as stated on page 6 (lines 134-135) in the research aims). This seems contradictory.

Finally, the authors expect autistic participants to increase social behaviours such as smiling in mixed neurotype dyads due to heightened social demands (H3c). What does heightened social demands refer to specifically?

I would suggest reformulating more clearly and consistently hypothesis 2 and 3 (and their sub-hypotheses).

Page 8, lines 185-186: The inclusion of self-diagnosed individuals in the autistic group should be mentioned upfront in the abstract. While I fully support including self-identified participants—given recruitment challenges and barriers to formal diagnosis—this information needs to be transparent from the outset. Group composition impacts result interpretation and generalizability and should also be discussed in the Discussion/Limitation section.

Related to this, I also wonder if it is appropriate to use “diagnostic status” as outcome variable name, given the mixed status of participants. Maybe a preliminary note would be useful to clarify what the authors mean by diagnostic status.

I'd also recommend reporting the distribution of clinical versus self-diagnosis across dyad types, not just by individual group membership.

Page 8, line 189: Could the authors provide more information on how dyad partners were matched. Did they try to match them by age or sex/gender ?

Page 11, line 254: What were the causes of coding disagreement for the variable “Smiling” ? Were they systematic or random ? This result should also be mentioned in the Discussion or Limitation section.

Typos

Page 5, 99-99: To address this gap, the Actor-Partner Interdependence Model (APIM) is a is a statistical model used to analyse data from dyadic relationships which allows for the simultaneous examination of how each person's behaviours and characteristics influence not only their outcomes but those of their partner.

Page 12, line 276: recording  recoding

6. PLOS authors have the option to publish the peer review history of their article (what does this mean?). If published, this will include your full peer review and any attached files.

Reviewer #1: No

Reviewer #2: No

---

## [Author Response · Author response to Decision Letter 1]

1 Jul 2025

25th June 2025

Re: Diagnostic status influences rapport and communicative behaviours in dyadic interactions between autistic and non-autistic people. Themis N. Efthimiou, Stephanie Lewis, Sarah J. Foster, Charlotte E.H. Wilks, Michelle Dodd, Lorena Jiménez-Sánchez, Danielle Ropar, Robert A. Ackerman, Noah J. Sasson, Sue Fletcher-Watson, and Catherine J. Crompton

Dear Dr. Pisula,

Thank you for your letter and for the opportunity to revise our manuscript for publication in PLOS ONE. We are grateful to you and the reviewers for the insightful and constructive feedback. We believe that by addressing these comments, we have significantly improved the manuscript.

Below, we provide a point-by-point response (in blue) to the comments from the journal, the editor, and both reviewers. We would like to clarify that the funders had no role in study design, data collection and analysis, decision to publish, or preparation of the manuscript, as indicated within the manuscript on page 32, lines 909-913. However, the foundation offered additional funding to allow us to include a condition in which participants were uninformed about the diagnostic status of their interaction partner. All changes in the manuscript have been marked using the 'Track Changes'.

We hope that the revisions and our responses are satisfactory and that the manuscript is now suitable for publication in PLOS ONE. We look forward to hearing from you.

Sincerely,

Themis Efthimiou, on behalf of all the authors.

Editor's Comments

1. General Comment: "Several aspects require further refinement and clarification. These include details regarding the research methodology, such as the formulation of hypotheses, the rationale behind the choice of indices, and a description of the process for matching dyad partners, as well as certain unclear sections in the introduction and discussion, as highlighted in the reviews."

Response: We have comprehensively revised the manuscript to address these points. Specifically, we have reformulated the hypotheses for clarity, provided a detailed rationale for our chosen indices, expanded the description of the dyad matching process, and clarified the noted sections in the Introduction and Discussion. Detailed responses to the specific reviewer comments on these points are provided below.

Reviewer Comments

Reviewer #1: The manuscript in question tackles an intriguing question using a novel and appropriate approach. It was also enjoyable to read. The authors are correct that this area is currently understudied and I believe their report will present a valuable addition to the study of Autism and Social Cognition. Nonetheless, I have a few thoughts I would like to offer for consideration before I can recommend this article for publication.

Response: We thank the reviewer for their time and feedback on our manuscript.

The introduction is very informative and focused, but I'd like to see a brief explanation/definition of the terms rapport and backchanneling before these are used in the argument - the former is sometimes ambiguous and can be operationalised in different ways, whereas the latter may not be known to many readers. The role of kinematics and why they matter when exploring rapport and comunication could also be established more as that is another non-intuitive factor.

Response: We agree that defining these key terms improves the manuscript’s clarity. Accordingly, we have revised the definition of ‘rapport’ in the introduction for greater precision. On page 3, lines 43-46, the text now states: “Social interactions are important for developing rapport, defined as a state of harmonious connection and mutual understanding between individuals[1,2]. This connection is developed through the intricate interplay of verbal and nonverbal communication and is crucial for forming social bonds and enhancing well-being[3–9].”

We have also added a definition for 'backchanneling' on page 4, lines 101-102, the text now includes the following: “...examining markers of rapport (mutual gaze and backchanneling, i.e., verbal and non-verbal cues like nodding or saying ‘uh-huh’ to show engagement)...”

I would also appreciate if the authors could briefly cover the deficits-or-differences argument, as this is very relevant here. The authors already bring up the current implicit understanding of Autism in terms of social deficits (line 59) that contrasts with the cited literature showing effective communication and high rapport among matched pairs. I would just like to see this addressed explciitly.

Response: Thank you for this suggestion. We have added the following to page 3, lines 61-64, which now reads: ‘Over recent years, autistic social traits have been increasingly interpreted as being “different” rather than “deficient”; that is, that autistic social communication is a natural variation in interactive styles rather than something that needs to be corrected or mitigated [29]. ‘

The authors mention larger group setting (line 92) and groups larger than dyads (line 114) - briefly addressing dyadic vs group interactions may be informative and clarify the scope of the current manuscript.

Response: We thank the reviewer for this important point about clarifying the manuscript's scope. We agree that the mention of larger groups could be confusing. To address this, we have made two changes: 1. On page 5 lines 107-115, we have removed the phrase “particularly in larger group setting” to keep the focus clearly on dyadic interactions. It now reads: “This finding further highlights the unique dynamics present in same-neurotype interactions, compared to mixed-neurotype interactions[10]. Together, these studies emphasise the need for deeper investigation into the interactional nuances within mixed and matched neurotype pairs.”

Where the use of APIM in various contexts is mentioned (line 114), we have added a sentence to reinforce the scope of the current study and justify our focus on dyads on page 6 lines 146-149, which now reads: “As dyadic interactions are the foundational building blocks of group dynamics, the present study focuses on two-person pairings to isolate these core interpersonal behaviours before they are compounded by the complexities of a group setting.”

For understandable reasons, the introduction and article in general treat "Autistic" and "Non-Autistic" as homogenous groups. It is okay and expected if intra-group diversity is not within scope, but this is a broad issue within the field at large so it should be explicitly stated as a limitation: Both co-occuring conditions, and different levels of Autism (certainly level 3) - would no-doubt affect the results and dyadic communication, as would other individual differences and conditions among the non-Autism group that were not recorded.

Response: We absolutely agree that both “autistic” and “non-autistic” are heterogenous, diverse groups. Autistic diagnostic status is just one of many factors that may affect how someone interacts and behaves. However, these were the comparison groups of interest in this study, and so this was our focus in the write up. We agree that intellectual disability, in either autistic or non-autistic participants, may affect results, and have now included in the limitations section on page 23, lines 607-608: “Fourth, our participants had relative high IQs, and as such may not be representative of the wider population.”

I notice that participants were recruited from different countries. Could the authors please clarify whether each country contributed the sae proportion to each dyad-arrangement? It doesn't have to be equal between countries but it should be equal across arrangements. If any country contributed, for example, a majority of Autistic dyads but a minority of Non-Autistic dyads, then cultural factors become an important consideration.

Response: Thank you for this important question. To ensure the recruitment site did not act as a confounding variable, we performed a Pearson's Chi-squared test of independence. The analysis showed no significant association between recruitment site and dyad type distribution (χ²(4) = 1.03, p = .91), indicating the proportions were not significantly different across sites. We have added a note to the manuscript to clarify this point. On page 8-9, lines 234-236, the text now states: ". To confirm that the distribution of dyad types was comparable across the three Universities, a Pearson's Chi-squared test of independence was performed. The analysis showed no significant association between recruitment site and dyad type (χ²(4) = 1.03, p = .91), indicating the groups were sufficiently balanced in this regard”. The raw counts are as follows:

Dallas: Autistic (n=12), Mixed (n=10), Non-autistic (n=12)

Edinburgh: Autistic (n=16), Mixed (n=12), Non-autistic (n=10)

Nottingham: Autistic (n=12), Mixed (n=12), Non-autistic (n=12)

Might the results regarding the diagnostic status of the partner be influenced by social desirability? It seems logical that one dyad member would be hesitant to report low rapport if they knew their partner was Autistic, which would explain the high rapport of mixed dyads. This would be an interesting effect in itself but would present a confound that should be mentioned. To further explore this, the authors could compare the rapport ratings within mixed pairs, looking for differencxes between the Autistic and non-Autistic partner's ratings.

I may be reading this wrong (which may affect my previous point), but in the paragraph starting on line 362, the authors seem to say that informed mixed dyads report higher rapport, which is driven by the Autistic member who, when informed, reports lower rapport. This is obviously contradictory so I am guessing something may have gotten switched around here or the paragraph is not entirely clear. Upon reviewing figure 3 and table 4, I think there is a mistake in line 363, which possibly should say that uninformed mixed dyads reported higher rapport, not lower. In that case my previous point becomes moot but I would still like to see a deeper discussion of this effect.

Response: Here we respond to the two previous comments. Thank you for your careful review and for identifying this error. You are absolutely correct - there was a mistake in our original text. The corrected statement should read that uninformed mixed dyads reported higher rapport than informed mixed dyads, not the reverse as we mistakenly wrote. We have made this correction on pages 18-19, lines 406-506.

The accurate interpretation is that when mixed dyads are uninformed of diagnostic status, they report significantly higher rapport (Mdiff = -81.1, p = .015, Cohen's d = -1.16), and as you correctly noted from our descriptive statistics, this effect appears to be driven by the autistic member who reports substantially higher rapport when uninformed (M = 447) compared to informed (M = 365.82) of their partner's neurotypical status.

This correction actually strengthens rather than undermines the interpretation, as it aligns with our central finding that autistic individuals are more sensitive to diagnostic awareness than to neurotype composition per se. The pattern now shows that autistic individuals benefit from non-disclosure when interacting with neurotypical partners, but benefit from disclosure when interacting with other autistic individuals - suggesting different optimal strategies depending on interaction context rather than a simple social desirability bias.

Regarding your social desirability concern, while this remains a theoretical possibility, the corrected pattern actually argues against this interpretation. If social desirability were the primary driver, we would expect consistently higher rapport ratings when diagnostic status is known (to avoid appearing prejudiced). Instead, we observe a more complex interaction where diagnostic awareness helps in some contexts (autistic dyads) but hinders in others (mixed dyads), recommending genuine differences in social experience rather than response bias.

We appreciate your thorough review, which helped us identify and correct this important error while clarifying the theoretical implications of our findings.

It is already part of the data but a direct test comparing ratings of Autistic participants, depending on whether they interacted with an Autistic or non-Autistic partner, may be interesting. Currently this is somewhat hard to judge due to mixed pairs being averaged across the two participants.

Response: We thank the reviewer for this suggestion. We conducted a direct comparison of rapport ratings from autistic participants in same-neurotype dyads versus those in mixed-neurotype dyads. An independent-samples t-test revealed no significant difference in self-reported rapport between these two groups, t(33.02) = 0.38, p = .71. This suggests that for autistic participants in our sample, their perception of rapport was not significantly influenced by the neurotype of their interaction partner. We have added this analysis to the manuscript.

Minor comments:

On line 96, the words "is a" are repeated.

Response: We thank the reviewer for spotting this error; it has been corrected.

Hypotheses H2b (line 152) and H3c (line 160) are phrased in a way that is very hard to parse. Please consider rewriting.

Response: Thank you for this feedback. We agree that the phrasing of H2b and H3c could be clearer. To improve readability while adhering to our pre-registered analysis plan, we have rewritten the sentences to make the predicted effects easier to understand without changing their substance.

The revised text aims to more clearly articulate the predicted partner effects (H2b) and the interaction effects (H3c) on participant behaviour. The revised text now reads:

“Second, we predict that interactional indices will mediate the relationship between diagnostic status and rapport (H2). Specifically, (H2a) we predict that autistic individuals will display reduced behavioural indices (e.g. smiling) and high levels of kinematics indices (i.e., less smooth movements), leading to lower perceived rapport. We also predict a partner effect (H2b), where an actor’s rapport is influenced by their partner's behaviour; we hypothesise that an autistic partner will exhibit decreased social behaviours and altered kinematics, which in turn will reduce the actor’s perceived rapport. Finally (H2c), we predict that the shared diagnostic status of the dyad moderates these mediation effects. We suggest that the influence of social and kinematic cues on rapport will be strongest in non-autistic dyads and weakest in autistic dyads, with mixed-neurotype dyads falling in between.

Finally, we hypothesise that autistic participants will exhibit diminished behavioural indices and higher kinematic indices, than non-autistic participants (H3a), with these interactional indices further influenced by their partner’s diagnostic status (H3b). Finally (H3c), we predict an interaction between the diagnostic statuses of the dyad members will influence interactional indices. Specifically, we hypothesise that autistic participants will exhibit more interactional indices in mixed-neurotype dyads compared to when interacting with an autistic partner. This may be a response to the heightened social demands of masking or adapting their natural interaction style to align with perceived neurotypical norms. Full details and mediation models are provided in Supplemental Materials S2.”

In hypothesis H3a (line 158), I believe the authors mean to say that Autistic participants will exhibit dimished or reduced multimodal indices, as the amount of indices stays the same.

Response: Thank you for this important and precise suggestion. You are correct; our original wording was imprecise. We intended to predict a difference in the values of the indices, not the number of indices measured. We have revised the text to use the term "diminished" for greater clarity, which better reflects the intent of our pre-registered hypothesis.

Errant ")" on line 190.

Response: We thank the reviewer for spotting this error; it has been corrected.

Reviewer #2: Thank you for the opportunity to review this article. This is a well-written paper presenting a thoughtfully designed study that makes a valuable contrib

---

## [Decision Letter · Decision Letter 1]

29 Jul 2025

Diagnostic status influences rapport and communicative behaviours in dyadic interactions between autistic and non-autistic people

PONE-D-25-10256R1

Dear Dr. Efthimiou,

We’re pleased to inform you that your manuscript has been judged scientifically suitable for publication and will be formally accepted for publication once it meets all outstanding technical requirements.

Kind regards,

Ewa Pisula

Academic Editor

PLOS ONE

Additional Editor Comments (optional):

Reviewers' comments:

Reviewer's Responses to Questions

**Comments to the Author**

1. If the authors have adequately addressed your comments raised in a previous round of review and you feel that this manuscript is now acceptable for publication, you may indicate that here to bypass the “Comments to the Author” section, enter your conflict of interest statement in the “Confidential to Editor” section, and submit your "Accept" recommendation.

Reviewer #1: All comments have been addressed

Reviewer #2: All comments have been addressed

2. Is the manuscript technically sound, and do the data support the conclusions?

Reviewer #1: Yes

Reviewer #2: (No Response)

3. Has the statistical analysis been performed appropriately and rigorously? 

Reviewer #1: Yes

Reviewer #2: (No Response)

4. Have the authors made all data underlying the findings in their manuscript fully available?

Reviewer #1: Yes

Reviewer #2: (No Response)

5. Is the manuscript presented in an intelligible fashion and written in standard English?

Reviewer #1: Yes

Reviewer #2: (No Response)

6. Review Comments to the Author

Reviewer #1: I thank the authors for thoroughly addressing all my comments and thoughts. I have no further concerns regarding this manuscript and recommend publication. It was an enjoyable, interesting read and I wish the authors good luck in the future.

Reviewer #2: (No Response)

7. PLOS authors have the option to publish the peer review history of their article (what does this mean?). If published, this will include your full peer review and any attached files.

Reviewer #1: **Yes: **Tim Vestner

Reviewer #2: No

---

## [Editor Report · Acceptance letter]

PONE-D-25-10256R1

PLOS ONE

Dear Dr. Efthimiou,

I'm pleased to inform you that your manuscript has been deemed suitable for publication in PLOS ONE. Congratulations! Your manuscript is now being handed over to our production team.

Kind regards,

on behalf of

Dr. Ewa Pisula

Academic Editor

PLOS ONE